# Baseline Expression of Exosomal miR-92a-3p and miR-221-3p Could Predict the Response to First-Line Chemotherapy and Survival in Metastatic Colorectal Cancer

**DOI:** 10.3390/ijms241310622

**Published:** 2023-06-25

**Authors:** Alexandra Gherman, Loredana Balacescu, Calin Popa, Calin Cainap, Catalin Vlad, Simona S. Cainap, Ovidiu Balacescu

**Affiliations:** 111th Department of Medical Oncology, University of Medicine and Pharmacy “Iuliu Hatieganu”, 34–36 Republicii Street, 400015 Cluj-Napoca, Romania; allexandragherman@gmail.com (A.G.); loredana_balacescu@yahoo.com (L.B.); 2Department of Medical Oncology, The Oncology Institute “Prof. Dr. Ion Chiricuta”, 34–36 Republicii Street, 400015 Cluj-Napoca, Romania; 3Department of Genetics, Genomics and Experimental Pathology, The Oncology Institute “Prof. Dr. Ion Chiricuta”, 34–36 Republicii Street, 400015 Cluj-Napoca, Romania; 4“Prof. Dr. Octavian Fodor” Regional Institute of Gastroenterology and Hepatology Cluj-Napoca, 19–21 Croitorilor Street, 400162 Cluj-Napoca, Romania; calinp2003@yahoo.com; 5Department of Surgery, Surgery Unit No 3, University of Medicine and Pharmacy “Iuliu Hațieganu” Cluj-Napoca, 19–21 Croitorilor Street, 400162 Cluj-Napoca, Romania; 6Department of Surgery, The Oncology Institute “Prof. Dr. Ion Chiricuta”, 34–36 Republicii Street, 400015 Cluj-Napoca, Romania; catalinvlad@yahoo.it; 7Department of Oncology, “Iuliu Hatieganu” University of Medicine and Pharmacy, 8 Victor Babes Street, 400012 Cluj-Napoca, Romania; 8Department of Mother and Child, Pediatric Cardiology, University of Medicine and Pharmacy “Iuliu Hatieganu”, 19–21 Croitorilor Street, 400162 Cluj-Napoca, Romania; cainap.simona@umfcluj.ro; 9Department of Paediatric Cardiology, Pediatric Clinic No 2, Emergency County Hospital for Children, 68 Motilor Street, 400370 Cluj-Napoca, Romania

**Keywords:** metastatic colorectal cancer, chemotherapy, response, survival, biomarkers, microRNA

## Abstract

The status of predictive biomarkers in metastatic colorectal cancer is currently underdeveloped. Our study aimed to investigate the predictive value of six circulating exosomal miRNAs derived from plasma (miR-92a-3p, miR-143-3p, miR-146a-5p, miR-221-3p, miR-484, and miR-486-5p) for chemosensitivity, resistance patterns, and survival. Thirty-one metastatic colorectal cancer patients were selected before receiving first-line irinotecan- or oxaliplatin-based chemotherapy. Blood samples were harvested at baseline and 4–6 months after the initiation of chemotherapy. The levels of exosomal expression for each miRNA were analyzed by qPCR. Our results for patients receiving first-line FOLFOX showed significantly higher baseline levels of miR-92a-3p (*p* = 0.007 **), miR-146a-5p (*p* = 0.036 *), miR-221-3p (*p* = 0.047 *), and miR-484 (*p* = 0.009 **) in non-responders (NR) vs. responders (R). Of these, miR-92a-3p (AUC = 0.735), miR-221-3p (AUC = 0.774), and miR-484 (AUC = 0.725) demonstrated a predictive ability to discriminate responses from non-responses, regardless of the therapy used. Moreover, Cox regression analysis indicated that higher expression levels of miR-92a-3p (*p* = 0.008 **), miR-143-3p (*p* = 0.009 **), miR-221-3p (*p* = 0.016 *), and miR-486-5p (*p* = 0.019 *) at baseline were associated with worse overall survival, while patients expressing higher baseline miR-92a-3p (*p* = 0.003 **) and miR-486-5p (*p* = 0.003 **) had lower rates of progression-free survival. No predictive values for candidate microRNAs were found for the post-chemotherapy period. In line with these findings, we conclude that the increased baseline exosomal expression of miR-92a-3p and miR-221-3p seems to predict a lack of response to chemotherapy and lower OS. However, further prospective studies on more patients are needed before drawing practice-changing conclusions.

## 1. Introduction

Colorectal cancer (CRC) is a significant cause of cancer-related morbidity and mortality worldwide, currently ranking the fourth most diagnosed and the third most deadly neoplasia [1]. The prognosis remains unfavorable in a metastatic setting, with a 5-year survival rate of 13.1% [2].

The role of biomarkers in cancer is essential not only in the initial diagnosis of the disease or at recurrence but also in the prognosis and prediction of responses to various therapies, allowing for the selection of an individualized therapeutic approach. In patients with a good performance status, the first-line treatment of metastatic colorectal cancer (mCRC) consists of chemotherapy doublets/triplets based on fluoropyrimidines, oxaliplatin, and/or irinotecan, to which a molecular targeted therapy is commonly added. Depending on the location of the primary tumor in the colon and the RAS gene’s mutational status, monoclonal anti-EGFR or anti-VEGF antibodies are chosen as appropriate [3]. However, predicting treatment responses in metastatic CRC still represents a significant challenge. Although several genetic alterations, such as the mutation status of KRAS and BRAF genes or microsatellite instability (MSI), represent prediction factors for CRC progression, they are not sufficient for predicting treatment responses in metastatic CRC. Investigating microbiota could also benefit CRC prognosis [4] and its metastasis [5,6]. Nevertheless, it is not feasible to use microbiota as a predictive factor for treatment responses in metastatic CRC given the complexity of investigating microbiomes as an aspect of personalized medicine.

Aside from the above-mentioned predictive factors for the response to therapy, there are currently no other validated clinical/biological/molecular factors for predicting the response to treatments and/or survival of these patients [7]. Identifying new molecular biomarkers, such as miRNAs, could represent the beginning of a paradigm shift in the therapeutic approach to mCRC. MiRNAs are small molecules of RNA of about 18–24 nucleotides; they are not involved in coding genetic information but play a crucial role in the post-transcriptional control of gene expression. They have been shown to be involved in most steps of carcinogenesis, functioning either as an oncomiR by targeting tumor-suppressing mRNA or as a tumor-suppressor miRNA by targeting oncogenes [8]. In addition to the oncogenic and tumor-suppressor roles, miRNAs act as mediators of inflammation, sustaining carcinogenesis. While inflammatory stimuli can alter the expression of specific miRNAs, some miRNAs mediate inflammation. Many biological processes are affected by inducing or repressing the activity of miRNA, and pro- or anti-inflammatory stimuli are generated [9]. The advantage of plasma miRNAs is that they require only a low volume of blood, are extremely stable, and can provide complex molecular information [10]. Previous data indicate that specific panels of serum miRNAs could represent prognostic markers for CRC, comparable with traditional markers, including carcinoembryonic antigen (CEA) and carbohydrate antigen 19-9 (CA19-9), or for the detection of CRC patients with distant metastasis [11]. Other approaches, including exploring the role of mast cells or the interaction of miRNAs with other biomarkers such as KRAS, are helpful in predicting treatment responses to classical cytotoxic chemotherapy or targeted therapy [12].

Most of the miRNAs that are detectable in serum/plasma are concentrated in exosomes. Valadi and colleagues were the first team to show the transfer of miRNA through exosomes to transport genetic information [13]. The transmembrane transportation of biological material is an essential element of homeostasis. Many normal cells and tumors secrete exosomes, the primary role of which is facilitating intercellular communication. In cancer, exosomes contribute to cell proliferation, invasion, and metastasis, escape from the immune system, angiogenesis [14], and chemoresistance. Exosomes prevent the inactivation of extracellular miRNA so that they can safely reach their recipient cells [15]. They have been detected in the blood, saliva, malignant ascites, breast milk, urine, cerebrospinal fluid, etc. Some studies have reported that malignant cells secrete more exosomes than normal ones, and patients with neoplasia have more circulating exosomes than healthy persons [16]. Current research focuses on their use as diagnostic, prognostic, and predictive biomarkers for therapies and their manipulation to treat various human pathologies. The costs incurred by their potential large-scale use could be reduced through the better selection of patients for expensive therapies.

To date, numerous preclinical and clinical studies have been conducted to identify and validate specific miRNAs as predictive biomarkers in metastatic colorectal cancer, yet none has been implemented for wide-scale usage in clinical practice. In this context, considering the importance of this first-line treatment option in managing patients with metastatic CRC, our study focused on evaluating the predictive role of six miRNAs of interest for assessing treatment with FOLFOX and FOLFIRI when used as a first-line therapeutic option in treating metastatic colorectal cancer.

## 2. Results

From January 2015 until December 2016, 31 patients with mCRC, who were treatment-naïve for metastatic disease, were included in this study. Baseline (B) and post-chemotherapy (PC) plasma samples (4–6 months after starting first-line chemotherapy) were available for all patients. The clinical follow-up of the patients lasted until 31 December 2019.

### 2.1. Patient and Tumor Characteristics

The demographic and clinicopathological characteristics of the included patients are described in Table 1. Out of the thirty-one patients included, seventeen received first-line chemotherapy with irinotecan and fluoropyrimidine (FOLFIRI); the remaining fourteen received chemotherapy with oxaliplatin and a fluoropyrimidine (FOLFOX). Sixteen patients had targeted therapy added to 1st line chemotherapy protocol between the two timepoints (baseline and postchemotherapy) according to the RAS mutational status (bevacizumab in twelve patients, cetuximab in 4 patients). Two patients achieved a complete response and fifteen a partial response; nine patients exhibited stable disease and the remaining five had progressive disease.

### 2.2. Investigating the Exosomal miRNAs of Interest

We first evaluated the signal amplification for the selected miRNAs in case–control cohorts, as follows: FOLFIRI (B), FOLFIRI (PC), FOLFOX (B), and FOLFOX (PC). Because all of the miRNAs had less than 33 CT amplification in the pooling groups (Table 2), they were further considered for individual assessment for the samples included in each group.

### 2.3. Predictive Role of Exosomal miRNAs for Chemotherapy Response to First-Line Chemotherapy in mCRC

To identify the associations between the miRNAs of interest and the response to first-line chemotherapy, we investigated the expression of these microRNAs in 17 patients treated with the FOLFIRI (FIRI) protocol and 14 patients with the FOLFOX (FOX) protocol at 2 different time points (baseline and post-chemotherapy). No statistically significant differences in microRNA expression between the two chemotherapy protocols were observed at baseline (B) and post-chemotherapy (PC). Our data suggest that there is no risk of associating false-positive expression levels with the response to therapy (Figure 1).

Furthermore, we tested the association of the miRNAs of interest with the response to therapy, regardless of the chemotherapy combination (FOLFOX or FOLFIRI) but according to the drugs used. We found increased baseline levels of exosomal miR-92a-3p (FR = 1.60, *p* = 0.047 *) and miR-484 (FR = 2.01, *p* = 0.015 *) in non-responders compared to responders, regardless of the chemotherapy protocol administered (Figure 2A), but no significant differences were observed in the post-chemotherapy samples (Figure 2B). We also noticed elevated levels of miR-221-3p (FR = 2.7) at baseline in the plasma exosomes of NR patients treated with FOLFOX or FOLFIRI, with a marginal *p*-value of 0.054 (Figure 2A).

Considering only patients treated with the FOLFOX protocol, our results indicated the significant upregulation of miR-92a-3p (FR = 2.17, *p* = 0.007 **), miR-146a-5p (FR = 3.32, *p* = 0.036 *), miR-221-3p (FR = 3.85, *p* = 0.047 *), and miR-484 (FR = 2.62, *p* = 0.009 **) expression at baseline in non-responders vs. responders (Figure 3A). Post-chemotherapy, there was no significant association between the exosomal miRNA expression and the treatment response (Figure 3B). For the FOLFIRI group, the expression of the miRNAs of interest exhibited no significant difference between the non-responders’ samples and the responders’ samples, either at baseline or post-chemotherapy.

We performed ROC analysis to assess the accuracy of the candidate microRNAs to distinguish between responses and non-responses at baseline and post-chemotherapy. The results suggest that the baseline values of miR-92a-3p (AUC = 0.735, 95% CI: 0.552–0.919, *p*-value = 0.026), miR-484 (AUC = 0.725, 95% CI: 0.527–0.924, *p*-value = 0.042), and miR-221-3p (AUC = 0.774, 95% CI: 0.593–0.955, *p*-value = 0.011) had the predictive ability to discriminate non-responders from patients who exhibited a response to therapy, while the post-chemotherapy values showed suboptimal prediction accuracy (Figure 4).

### 2.4. Predictive Role of Exosomal miRNAs for Progression-Free Survival and Overall Survival in mCRC

These miRNAs’ ability to predict progression-free survival during first-line chemotherapy (PFS1) and overall survival (OS) was further evaluated. At the time of evaluation, the median PFS1 was 17 months, and the median OS was 22 months.

The univariate COX regression analysis (UA) revealed the statistically significant association of PFS1 with baseline values of miR-486-5p (*p* = 0.003 **) and miR-92a-3p (*p* = 0.003 **), showing that the mCRC patients with high levels of baseline exosomal expression of miR-486-5p or miR-92a-3p had significantly shorter PFS than those with low baseline exosomal levels of these miRNAs (Table 3). Although miR-484, miR-146a-5p, and miR-143-3p 5p had low *p*-values in UA, they did not reach the significance threshold (Table 3). Considering the OS parameter, we found that increased baseline exosomal expression of miR-486-5p (*p* = 0.019 *), miR-92a-3p (*p* = 0. 008 **), miR-221-3p (*p* = 0.016 *), and miR-143-3p (*p* = 0.009 **) is predictive for a worse OS in mCRC patients (Table 3). MiRNAs with predictive value (*p* < 0.05) in univariate analysis were included in the multivariate analysis (MA) of survival, but none of the miRNAs showed independent predictive value for PFS1 and OS (Table 3). Additionally, the analysis of the post-chemotherapy data did not confirm that any of the studied miRNAs had an ability to predict PFS1 and OS (Table 3).

## 3. Discussion

Although genetic changes play a major role in CRC development and its progression, epigenetic alterations such as DNA methylation or miRNA alteration also contribute to the validation of the cancer phenotype. The role of miRNAs in CRC development, invasion, and metastasis has been well documented [7,17]. However, the identification of biomarkers in a liquid biopsy, especially exosomal miRNAs, is still under investigation. Despite the numerous publications in the field, work is still underway to validate these results and implement them in clinical practice. Considering the results of our study, we were further interested in identifying the results of other research teams in predictive miRNAs in plasma exosomes in mCRC. However, due to the limited number of articles identified, we extended our literature search to include free-circulating miRNAs as potential biomarkers.

The miR-17-92p cluster has been extensively studied for its role in CRC tumorigenesis, metastasis, and responses to therapy. As a member of the miR-17-92 precursor cluster, miR-92a-3p is known to be an oncogenic miRNA in CRC. We showed that an increased baseline plasma exosome level of miR-92a-3p was associated with a lack of a therapeutic response, regardless of the chemotherapy protocol administered (*p* = 0.047), and also in the subgroup of patients who received FOLFOX chemotherapy (*p* = 0.007). Our data indicated that high levels of miR-92a-3p at baseline in plasma exosomes are associated with low OS (*p* = 0.008); additionally, patients with increased expression had lower PFS1 levels (*p* = 0.003). These data are supported by the results of other researchers, who aimed to investigate both the prognostic and predictive roles of miR-92a-3p. In 2019, Hu’s team [18] analyzed the levels of miR-92a-3p in the serum exosomes of patients with chemo-responsive CRC (*n* = 18) and chemo-resistant CRC (*n* = 18) in patients treated with FOLFOX. They showed that the level of miR-92a-3p was significantly higher in non-responders than in responders and concluded that miR-92a-3p in serum exosomes could be a good predictor of metastasis and chemoresistance in CRC. They also showed that the serum exosome level of miR-92a-3p was highest in patients with metastatic CRC vs. healthy subjects, emphasizing the diagnostic aspects and its prognostic role. Along the same lines, Fu et al. [19] studied the prognostic role of miR-92a-3p and miR-17-5p in mCRC, showing that the serum exosome levels of miR-92a-3p were significantly higher in metastatic patients than in those with localized disease (*p* ˂ 0.0001), thus being both a diagnostic and a prognostic biomarker. Matsumura and colleagues [20] included 124 patients diagnosed with mCRC, with or without tumor recurrence, and analyzed their serum samples in their study. The gene expression level of the miR-17-92a cluster in the exosomes was correlated with the relapse of the CCR. Poel et al. [21] conducted a study to identify miRNAs as biomarkers for sensitivity to palliative chemotherapy in mCRC. In total, 132 patients treated with 5-FU + Oxaliplatin ± Bevacizumab were prospectively included and evaluated for their response to treatment according to RECIST 1.1 criteria. Baseline tissue levels and the serum expression of miR-17-5p, miR-20a-5p, miR-30a-5p, miR-92a-3p, miR-92b-3p and miR-98-5p were evaluated. The serum expression levels of miR-92a-3p and miR-98-5p significantly improved the predictive value for the chemotherapy response of the clinical parameters studied. Conev et al. [22] investigated the predictive value of miR-17, miR-21, miR-29a, and miR-92 to diagnose relapse in patients with CRC who were treated by surgery and adjuvant chemotherapy. The serum expression of miR-17, miR-21, and miR-92 was significantly higher in patients experiencing disease relapse. Thus, in this study, the serum expression levels of the three miRNAs could diagnose cancer relapse in stage III CRC patients. It was found that miR-92a could be involved in CRC metastasis via a PTEN-mediated PI3K/AKT pathway [23]. Moreover, miR-92a can activate the Wnt/β-catenin pathway and inhibits mitochondrial apoptosis by directly targeting FBXW7 and MOAP1 and mediating 5-FU/L-OHP resistance in CRC [18].

Tumor growth and metastasis are inhibited by miR-143-3p, which is reported to be a tumor suppressor in CRC. Previous studies have shown that its expression is downregulated in CRC. Sahami-Fard et al. [24] evaluated the serum expression levels of miR-143-3p, miR-424-5p, miR-212-3p, and miR-34a-3p in patients with CRC. Their analysis showed an increased expression of miR-424-5p (*p* < 0.001) and decreased miR-143-3p (*p* < 0.001) in CRC patients; the low expression of miR-143-3p exhibited more aggressive tumor features. Romero-Lorca and colleagues [25] included 76 patients with mCRC in their study and showed that, in paraffin-embedded biopsies, the overexpression of miR-143-3p was associated with a significantly better PFS. However, our results show that an increased gene expression level of miR-143-3p at baseline was associated with low OS (*p* = 0.009). In line with our results, another clinical study [26] showed that, in mCRC patients treated with capecitabine as part of the CAIRO trial, patients with low expression of miR-143 in their primary tumor experienced a better PFS than those with high expression. According to the authors, one possible explanation for this finding relates to the involvement of FXYD3, a putative target of miR-143, which can affect some transporters involved in the uptake of fluoropyrimidines and, thus, the treatment response.

Our data indicate that an increased baseline level of plasma exosomal miR-146a-5p was associated with a lack of a therapeutic response in the subgroup of patients who received FOLFOX chemotherapy (*p* = 0.036). We did not find any studies reported in the literature that assess the role of circulating miR-146a-5p in patients with CRC. However, in a previous study, Lu et al. [27] demonstrated that high levels of miR-146a-5p in CRC are associated with cell migration and invasion via the carboxypeptidase M/src-FAK pathway.

The results of our study show that an increased baseline expression of the oncogenic miR-221-3p in plasma exosomes was associated with low OS (*p* = 0.016) and with a lack of a therapeutic response in the subgroup of patients who received FOLFOX chemotherapy (*p* = 0.047). The high expression of miR-221 in CRC tissues was closely associated with a shorter survival time and a high level of risk for CRC prognosis [28]. Moreover, by epigenetic reprogramming, an anti-metastatic effect was observed in trials with NP-encapsulated anti-miR-221 [29]. Ulivi and colleagues [30] analyzed the predictive role of free circulating miRNAs in patients with mCRC included in the ITACa study, treated by FOLFOX/FOLFIRI associated with an antiangiogenic treatment (Bevacizumab). In total, 52 patients treated according to the protocol were included. Of the miRNAs considered, miR-21-5p and miR-221-3pb were significantly related to the RAS mutational status, thus having a predictive role for the response to anti-EGFR agents. In a previous paper, Dokhanci et al. [31] revealed that exosomal miR-221-3p induces angiogenesis in vitro by regulating the STAT3/VEGFR-2 signaling axis by targeting SOCS3 in endothelial cells. Moreover, miR-221-3p upregulation predicts a poor overall survival rate by suppressing SPINT1 expression and activating the liver hepatocyte growth factor (HGF), which leads to the formation of a favorable premetastatic niche and CRC metastasis [32]. To our knowledge, no studies discuss the associations of exosomal miR-221-3p in the treatment predictions of metastatic CRC.

We pointed out that an increased baseline level of miR-484 in plasma exosomes was associated with lack of a therapeutic response, regardless of the chemotherapy protocol administered (*p* = 0.015), but also in the subgroup of patients who received FOLFOX chemotherapy (*p* = 0.009). Regarding the modulation of the expression induced by chemotherapy, there was an increased expression level from baseline to post-chemotherapy of 1.68 x in responders vs. non-responders. Kjersem et al. [33] studied the predictive potential of certain plasma-free miRNAs in predicting responses to FOLFOX chemotherapy. They analyzed the gene expression levels of selected miRNAs in plasma at baseline and after 4 cycles of chemotherapy in 24 patients with CRC, who were included in the NORDIC ACT clinical trial (12 responders and 12 non-responders). Then, the miRNA that indicated a differential expression between the two groups was validated in 150 patients. miR-106a, miR-484, and miR-130b were overexpressed in non-responders, with a significant differential expression at baseline. MiR-484 constitutes a tumor-suppressor miRNA and plays a role in the epithelial-to-mesenchymal transition (EMT) by targeting the DLK1 gene. [34].

The last miRNA investigated in our study was miR-486-5p. We observed that patients with high levels of miR-486-5p in the plasma exosomes at baseline have shorter PFS1 (*p* = 0.003) and OS (*p* = 0.019) than those with low baseline expression, so it represents a negative prognostic factor in mCRC. No significant associations have been established between the level of miR-486-5p and the response to therapy. Liu and colleagues [35] showed that, in patients with a localized stage of CRC, the level of miR-486-5p in the circulating exosomes decreases after surgery, which opens up new research directions regarding the potential value of predictive biomarkers for tumor resection or the complete tumor response to therapies. However, their results do not agree with those obtained by other teams. Bjørnetrø et al. [36] showed that, among other miRNAs, miR-486-5p is a hypoxia biomarker in locally advanced rectal cancer. At the time of diagnosis, they collected plasma samples from 24 patients with locally advanced rectal cancer and showed an association between low levels of miR-486-5p and miR-181a-5p in plasma exosomes and an invasive phenotype of the primary tumor (*p* = 0.029) and lymph node invasion (*p* = 0.024). There are few data related to the role of miR-486-5p in CRC metastasis. One suggested mechanism involves miR-486-5p promoting CRC proliferation and migration by activating the PLAGL2/IGF2/β-catenin signal pathway [35].

## 4. Materials and Methods

### 4.1. Study Design and Patients Included

Our study comprised a cohort study, comparing the metastatic subgroups with each other depending on their treatment response [37]. This prospective study included patients treated with first-line chemotherapy for metastatic colorectal cancer in a tertiary cancer center, the Oncology Institute “Prof. Dr. I. Chiricuta”, Cluj-Napoca, Romania; patients were enrolled between January 2015 and December 2016 and followed up until 31 December 2019. All patients agreed to be included in the study and signed the informed consent form. All data collected were obtained from the Institutional Cancer Registry. The study was approved by the Institutional Ethics Committee. Included patients were adults aged ≥ 18 years old with a histopathological diagnosis of colorectal cancer with either synchronous or metachronous metastases, and available data regarding demographics, clinical-pathological characteristics, and an adequate response to treatment evaluation. We excluded patients presenting multiple synchronous neoplasia and patients for whom the blood samples were suboptimal for further processing (i.e., macroscopical hemoglobin).

### 4.2. Treatments Administered and Response Evaluation

All patients received first-line chemotherapy for metastatic colorectal cancer, consisting of a fluoropyrimidine (5-Fluorouracil for most patients, or capecitabine) and irinotecan or oxaliplatin. In most cases, chemotherapy was accompanied by a targeted therapy (anti-EGFR or anti-VEGF), according to the tumor RAS mutational status and tumor characteristics. The response to therapies was evaluated every 2–3 months by CT scans and, in only a few cases, by MRI. According to the description of the CT scans and the RECIST 1.1 criteria, patients were considered responders if they had either a complete or partial response and non-responders if they had stable or progressive disease.

### 4.3. Plasma Collection

The whole blood for each patient was collected by venipuncture in an EDTA blood-collection tube in the same timeframe (8–12 a.m.). The samples were collected, at baseline, before the first administration of the first cycle of chemotherapy for metastatic disease, and at post-chemotherapy, 4–6 months after the initiation of chemotherapy in patients still receiving first-line chemotherapy, or before switching to second-line chemotherapy in patients who had progressed. All samples were collected before the administration of chemotherapy.

Each whole-blood sample was registered under a unique code and processed immediately under similar conditions. Plasma was separated from the whole-blood samples by double successive centrifugation at 4000 and 12,000 rpm for 10 min at 4 °C. The provided plasma was aliquoted (400 µL/tube) and stored at −80 °C until its further processing.

### 4.4. Isolation of Plasma-Derived Exosomes and RNA Extraction from Them

The isolation of plasma-derived vesicles was carried out using a specific commercial conditioning solution, *Total Exosome Isolation Kit from Plasma* (Thermo Fisher, Waltham, MA, USA). An additional plasma filtration using a 0.8 μm filter was included before the precipitation step to increase the quality of vesicles specific to the exosome domain. The filtered plasma was then treated with 0.05 *v*/*v* Proteinase K and incubated for 10 min at 37 °C. After incubation, 120 µL of conditioning solution was added to the plasma, followed by 30 min at 4 °C and 10,000× *g* centrifugation. The resulting precipitate (plasma-derived vesicles/exosomes) was resuspended in 200 μL PBS solution and used for RNA extraction, using the Total Exosome RNA and Protein Isolation Kit (Invitrogen, Carlsbad, CA, USA) according to the manufacturer’s protocol. By introducing a supplementary step of 0.8 μm filtration, we considerably improved the exosomes’ purity by maintaining the small EVs and by eliminating the large EVs when evaluating by transmission electron microscopy (TEM) and nanoparticle tracking analysis (NTA), as previously discussed [38]. The exosome denaturation was performed by adding a 2× denaturing solution, while the isolation of RNA was based on the classical method using an equivalent volume of acid-phenol: chloroform. After recovery, during the aqueous phase, the RNA was precipitated by 1.25 volumes of 100% ethanol and purified by passing it through a gel–silicate column. In addition to the protocol, in the extraction process, a volume of 3.4 μL of an exogenous control of miR-39 (2 × 10^8^ transcripts/extraction) was used for each sample. A final volume of 100 μL of exosome-derived RNA was provided for each sample and stored at −80 °C until further use. The QC of the extracted RNAs was evaluated using a NanoDrop ND-1000 spectrophotometer (NanoDrop Technologies, Wilmington, DE, USA).

### 4.5. Selection of miRNAs of Interest 

The available data regarding the predictive role of circulating exosomal miRNA for treatment responses in metastatic CRC are inadequate. Consequently, we conducted a literature study to identify new potential targets that could be investigated as predictive biomarkers for mCRC. First, we explored the specific tissue miRNAs involved in CRC development and metastasis [17,39]. Furthermore, we identified the common exosomal and tissue miRNAs for CRC, which could be considered predictive biomarkers for mCRC [7,40]. Additionally, we focused on the CRC circulating miRNAs associated with tumor progression and also with treatment prediction [41,42,43]. Our analysis revealed five miRNAs, miR-92a-3p, miR-146a-5p, miR-221-3p, miR-484, miR-486-5p, and miR-143-3p, as suitable to be investigated for treatment predictions for mCRC.

Because no optimal strategy is generally accepted for exosomal miRNA normalization, we choose a combination of endogenous and exogenous control miRNAs to increase the accuracy of the exosomal miRNA expression. Recent data suggest that using the same type of RNA species (miRNAs) as normalizers may be a more accurate strategy than using other species of RNA such as RNU6B small nuclear RNA (U6) [44]. Consequently, based on previous data related to exosomal miRNA normalization [45], we include miR-16-5p as an endogen normalizer and cel-miR-39 as spike-in (2 × 10^8^ transcripts) as an exogenous normalizer.

### 4.6. Assessment of the Expression of miRNAs of Interest

For the case–control study, we generated four pools of RNA aliquots, one for each group (FOLFIRI baseline (B), FOLFIRI post-chemotherapy (PC), FOLFOX baseline (B), and FOLFOX post-chemotherapy (PC)). We chose the one-step advanced miRNA system to evaluate the level of miRNA expression, considering that it has the advantage of enabling many miRNAs to be simultaneously assessed using the same pre-amplified cDNA. A volume of 4 µL of exosome-derived RNA for each sample was used for the cDNA synthesis with the TaqMan^®^ Advanced miRNA cDNA Synthesis Kit, according to the manufacturer’s recommendations (Thermo Fisher). Compared to the classical system, the advantage of the advanced miRNA system is that it includes a pre-amplification step for the entire spectrum of miRNA contained in the sample, proportionally for all mature miRNA species existing in the samples. After the pre-amplification, each cDNA was diluted by 1/10 with RNase-free water prior to evaluation by polymerization chain reaction (PCR). Furthermore, 2.5 μL of diluted cDNA was used to investigate the expression of miRNAs of interest with a TaqMan^®^ Fast Advanced Master Mix (2×) and specific miRNA advanced assays (Thermo Fisher Scientific) in a 10 µL reaction volume using a LightCycler (Roche Basel, Switzerland) 480 device under specific miR-advanced PCR conditions: activation of the UNG enzyme at 55 °C for 2 min and Taq polymerase activation at 95 °C for 20 s, followed by 40 PCR amplification cycles based on 2 amplification cycles at 3 s to 95 °C and 30 s to 60 °C, respectively. The expression level of the miRNA of interest was evaluated by ΔΔCT analysis of NR relative to R groups with the normalization of CT values at the difference between the two normalization controls, the endogenous one (miR-16-5p) and the exogenous one (cel-miR-39).

### 4.7. Statistical Analysis

According to the data distribution, the qPCR data were evaluated with the Mann–Whitney U test or the unpaired sample *t*-test for two categorical variables or the Kruskall–Wallis test, followed by Dunn’s multiple comparison post hoc test in the case of three categorical variables. The area under the ROC curve (AUC) was used to evaluate the microRNAs’ capacity to discriminate between responders and non-responders. The PFS1 was measured from the date of 1st line of chemotherapy to the date of 2nd line of chemotherapy or until the end of treatment in patients without disease progression. OS was calculated from the start of chemotherapy until the date of death or the last evaluation date. We used Cox proportional hazards regression models to evaluate the association between PFS1, respective OS, and miRNAs. For this purpose, the expression of each microRNA was dichotomized according to its median value, dividing the patients into two groups, with low and high expression.

## 5. Conclusions

Considering the small number of samples used, the data contained in this study should be regarded a hypothesis generator for future studies, given their limited statistical power for clinical applicability. Our data highlighted high baseline levels of exosomal miR-92a-3p, miR-146a-5p, miR-221-3p, and miR-484, which were correlated with lack of a response to FOLFOX chemotherapy, as well as the discriminatory power of miR-92a-3p, miR-221-3p, and miR-484 for non-responder patients regardless of the therapy used. Furthermore, increased exosomal levels of miR-92a-3p and miR-486-5p, evaluated at the onset of treatment, are associated with lower OS and PFS1, while increased exosomal levels of miR-143-3p and miR-221-3p at the beginning of therapy have a predictive value for shorter OS. Our results suggest that the increased baseline expression of exosomal miR-92a-3p and miR-221-3p predicts a lack of response to chemotherapy and a lower OS. However, the validation of these data on larger cohorts of advanced CRC patients will be able to confirm the role of these miRNAs in predicting a lack of response to FOX chemotherapy in clinical practice.

## Figures and Tables

**Figure 1 ijms-24-10622-f001:**
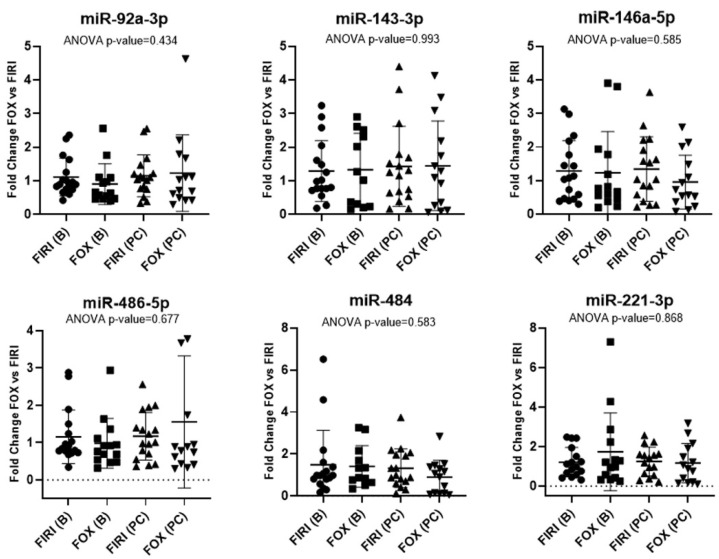
Fold change of the six miRNAs of interest (miR-92a-3p, miR143-3p, miR-146a-5p, miR-484, miR-486-5p, and miR-221-3p) in samples collected at baseline (B) and post-chemotherapy (PC) for the two therapeutic regimens administered: FOLFIRI (FIRI) and FOLFOX (FOX).

**Figure 2 ijms-24-10622-f002:**
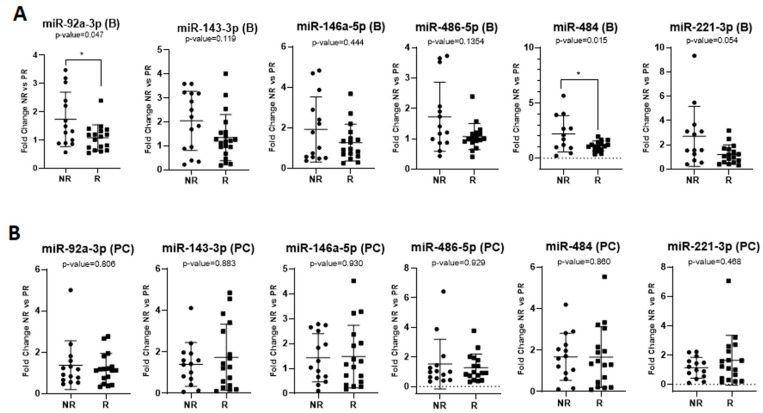
MicroRNA expression at baseline (B) presented in the (**A**) and post-chemotherapy (PC) presented in the (**B**), regardless of the therapy regimen administered in non-responder (NR) vs. responder (R) patients. The fold change for each sample was calculated relative to the PR group. * *p* ≤ 0.05.

**Figure 3 ijms-24-10622-f003:**
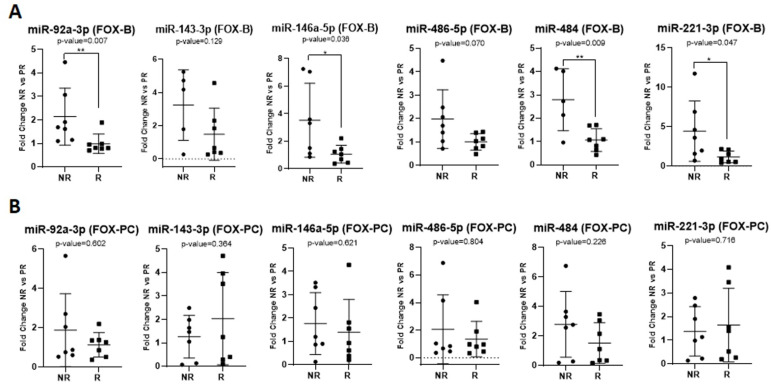
MicroRNA expression in the FOLFOX (FOX) therapy regimen at baseline (B) presented in the (**A**), and post-chemotherapy (PC) presented in the (**B**). The fold change for each sample was calculated relative to the PR group. * *p* ≤ 0.05; ** *p* ≤ 0.01.

**Figure 4 ijms-24-10622-f004:**
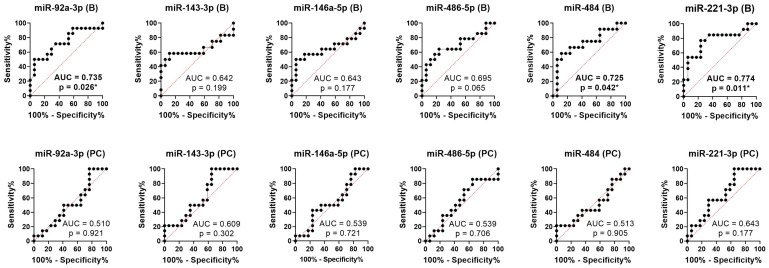
ROC curves for the investigated microRNAs at baseline (B) and post-chemotherapy (PC), regardless of the type of therapy. * *p* ≤ 0.05.

**Table 1 ijms-24-10622-t001:** Clinicopathological and chemotherapeutic data of the patients included in the study.

Variable	N = 31	
Age	Mean	59.35
	Median	61
Gender	Male	19 (61.29%)
	Female	12 (38.71%)
Tumor location	Right colon	10 (32.26%)
	Left colon	21 (67.74%)
Tumor grade	G1	4 (12.90%)
	G2	21 (67.74%)
	G3	6 (19.36%)
Metastases type	Synchronous	19 (61.29%)
	Metachronous	12 (38.71%)
Metastatic sites	single organ	16 (51.61%)
	multiple organs	15 (48.39%)
Chemotherapy protocol in first-line treatment	FOLFOX ^1^	14 (45.16%)
	FOLFIRI ^2^	17 (54.84%)

^1^ FOLFOX = oxaliplatin + 5 fluorouracil + folinic acid. ^2^ FOLFIRI = irinotecan + 5 fluorouracil + folinic acid/capecitabine.

**Table 2 ijms-24-10622-t002:** Data for the miRNAs of interest and their CT amplification values when evaluated in pooling interest groups: FOLFIRI (B), FOLFIRI (PC), FOLFOX (B), and FOLFOX (PC).

No.	miARN Name/ Accession Number	miRNA Code	FOLFIRI		FOLFOX	
			CT (B)	CT (PC)	CT (B)	CT (PC)
1	cel-miR-39 (MIMAT0000069)	uagcagcacguaaauauuggcg	18.29	18.07	18.67	18.27
2	hsa-miR-16-5p (MIMAT0000069)	uagcagcacguaaauauuggcg	25.88	26.02	26.15	26.76
3	hsa-miR-92a-3p (MIMAT0000092)	uauugcacuugucccggccugu	24.71	26.10	24.58	24.49
4	hsa-miR-143-3p (MIMAT0000435)	ugagaugaagcacuguagcuc	30.29	30.66	29.49	30.05
5	hsa-miR-146a-5p (MIMAT0000449)	ugagaacugaauuccauggguu	28.28	29.54	28.34	28.57
6	hsa-miR-486-5p (MIMAT0002177)	uccuguacugagcugccccgag	24.84	26.46	24.29	24.20
7	hsa-miR-484 (MIMAT0002174)	ucaggcucaguccccucccgau	29.11	29.28	28.05	29.09
8	hsa-miR-221-3p (MIMAT0000278)	agcuacauugucugcuggguuuc	26.46	27.18	25.94	26.06

**Table 3 ijms-24-10622-t003:** Univariate (UA) and multivariate (MA) Cox regression analysis for progression free survival during first-line chemotherapy (PFS1) and overal survival (OS), at baseline (B) and post-chemotherapy (PC), according to dichotomized miRNAs expression (High vs. Low). The hazard ratio (HR) was obtained from Cox proportional hazards regression models. HR > 1 indicates that the increased microRNA expression (High) is associated with poor survival. * *p* ≤ 0.05; ** *p* ≤ 0.01.

		miR-92a-3p *p*-Value	mir-143-3p *p*-Value	miR-146a-5p *p*-Value	miR-486-5p *p*-Value	miR-484 *p*-Value	mir-221-3p *p*-Value
**PFS1**							
**UA** ** *High* ** ** vs. *Low***	B	**0.003 **** **HR > 1**	0.06 HR > 1	0.07 HR > 1	**0.003 **** **HR > 1**	0.074 HR > 1	0.193 HR > 1
	PC	0.718 HR > 1	0.699 HR < 1	0.32 HR > 1	0.887 HR < 1	0.975 HR < 1	0.4848 HR < 1
**MA** ** *High* ** ** vs. *Low***	B	0.512 HR > 1	-	-	0.44 HR > 1	-	-
	PP PC CC	-	-	-	-	-	-
**OS**							
**UA** ** *High* ** ** vs. *Low***	B	**0.008 **** **HR > 1**	**0.009 **** **HR > 1**	0.112 HR > 1	**0.019 *** **HR > 1**	0.161 HR > 1	**0.016 *** **HR > 1**
	PC	0.097 HR > 1	0.428 HR > 1	0.452 HR > 1	0.181 HR > 1	0.19 HR > 1	0.601 HR > 1
**MA** ** *High* ** ** vs. *Low***	B	0.768 HR > 1	0.096 HR > 1	-	0.765 HR < 1	-	0.91 HR < 1
	PC	-	-	-	-	-	-

## Data Availability

Data is contained within the article.

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
