# Peer review of "Baseline Expression of Exosomal miR-92a-3p and miR-221-3p Could Predict the Response to First-Line Chemotherapy and Survival in Metastatic Colorectal Cancer"

_ijms, 2023, doi:10.3390/ijms241310622_

Round 1
Reviewer 1 Report
This paper aimed to investigate the predictive value of six circulating exosomal miRNAs derived from plasma for chemosensitivity, resistance patterns and survival in metastatic colorectal cancer patients. The study included 31 patients with mCRC, treatment-naïve for metastatic disease, who were included in the study from January 2015 until December 2016. Baseline and post-chemotherapy plasma samples were available in all patients. The levels of expression for each miRNA were analyzed by qPCR. Cox regression analysis was used to determine the association between miRNA expression and overall survival and progression-free survival.
Comments
1. It is important to note that the study had a small sample size of only 31 patients, which may limit the generalizability of the findings.
2. The study only investigated six specific miRNAs, and there may be other miRNAs that could be predictive of chemosensitivity and survival in metastatic colorectal cancer patients. Furthermore, the study only investigated the predictive value of miRNAs in patients receiving first-line irinotecan- or oxaliplatin-based chemotherapy, and the findings may not be applicable to patients receiving other types of chemotherapy.
3. The study did not discuss the mechanisms by which the miRNAs may be predictive of chemosensitivity and survival.
4. Absence of a control group or a comparison group, make it difficult to rely on the conclusions of this study. A control group is important to provide a basis for comparison and to help determine whether the results are due to the treatment being investigated or to other factors. Without a control group, it can be difficult to determine whether any observed effects are truly due to the treatment being investigated or to other factors such as chance, bias, or confounding variables.
5. A control group would have been used as well to validate the findings in this study.
6. Why ROC curves and AUC values were not used to evaluate the predictive value of the significant miRNAs being investigated in this study?
English language review is required to improve the quality of the manuscript.
Author Response
Dear Reviewer, thank you for your relevant comments. By answering your request, we think that our paper was drastically improved.
Sincerely,
Dr. Ovidiu Balacescu
Reviewer 2 Report
The discussion must be modified (and shortened) making it less of a static list
The authors should discuss also the other markers currently used:
- Unlocking the Potential of the Human Microbiome for Identifying Disease Diagnostic Biomarkers. Diagnostics (Basel). 2022 Jul 19;12(7):1742. doi: 10.3390/diagnostics12071742
- Mast Cells, microRNAs and Others: The Role of Translational Research on Colorectal Cancer in the Forthcoming Era of Precision Medicine. J Clin Med. 2020 Sep 3;9(9):2852. doi: 10.3390/jcm9092852
- Many Others
- Please add the limitations of the study
- Please add the clinical implications
Moderate English-language revision needed.
Author Response

(The authors gave the same response as above.)

Round 2
Reviewer 1 Report
Dear Authors,
It is very unfortunate that my comments were not addressed at all. Please see attached file.

There is no consistence in the language used and the proper flow of arguments in the current reviewed version of the manuscript. This should be improved more.
Author Response
Dear Reviewer, I am trying to understand why the report, including our answers to your questions, wasn't uploaded!
I've attached one again our answers point-by-point to your requests. Thank you for your questions; by answering your request, we drastically improved our paper.
Sincerely,
Dr. Ovidiu Balacescu

Reviewer 2 Report
I think the role of miRNA must be further highlighted:
- MicroRNAs in the prognosis and therapy of colorectal cancer: From bench to bedside. World J Gastroenterol. 2018 Jul 21;24(27):2949-2973. doi: 10.3748/wjg.v24.i27.2949
Mast Cells, microRNAs and Others: The Role of Translational Research on Colorectal Cancer in the Forthcoming Era of Precision Medicine. J Clin Med. 2020 Sep 3;9(9):2852. doi: 10.3390/jcm9092852
Minor
Author Response
Dear Reviewer, please find in the attached report our answer to your requests.
Sincerely,
Dr. Ovidiu Balacescu
